# Diagnosis, Management, and Therapy of Fetal Ovarian Cysts Detected by Prenatal Ultrasonography: A Report of 36 Cases and Literature Review

**DOI:** 10.3390/diagnostics11122224

**Published:** 2021-11-28

**Authors:** Takeya Hara, Kazuya Mimura, Masayuki Endo, Makoto Fujii, Tatsuya Matsuyama, Kazunobu Yagi, Yoko Kawanishi, Takuji Tomimatsu, Tadashi Kimura

**Affiliations:** 1Department of Obstetrics and Gynecology, Graduate School of Medicine, Osaka University, Suita 565-0871, Japan; tttake.0303@gmail.com (T.H.); endo@gyne.med.osaka-u.ac.jp (M.E.); matsu-tatsuya@muh.biglobe.ne.jp (T.M.); kamisama3121984@yahoo.co.jp (K.Y.); angeltears90@hotmail.co.jp (Y.K.); tomimatsu@gyne.med.osaka-u.ac.jp (T.T.); tadashi@gyne.med.osaka-u.ac.jp (T.K.); 2StemRIM Institute of Regeneration-Inducing Medicine, Osaka University, Suita 565-0871, Japan; m.fujii@sahs.med.osaka-u.ac.jp; 3Department of Children’s and Women’s Health, Division of Health Science, Graduate School of Medicine, Osaka University, Suita 565-0871, Japan

**Keywords:** fetal diagnosis, fetal therapy, fetal ovarian cyst, in-utero aspiration

## Abstract

Background: Fetal ovarian cysts are the most frequently diagnosed intra-abdominal cysts; however, the evidence for perinatal management remains controversial. Methods: We retrospectively reviewed cases of fetal ovarian cysts diagnosed by prenatal ultrasonography at our institution between January 2010 and January 2020. The following were investigated: gestational age at diagnosis, cyst size, appearance, prenatal ultrasound findings, and postnatal outcomes. Prior to 2018, expectant management was applied in all cases; after 2018, in utero aspiration (IUA) of simple cysts ≥40 mm was performed. Results: We diagnosed 29 and seven simple and complex cysts, respectively. Fourteen patients had simple cysts with a maximum diameter <40 mm, and two of them progressed to complex cysts during follow-up; however, when the diameter was limited to <35 mm, no cases showed progression to complex cyst. Fifteen of the simple cysts were ≥40 mm; three progressed to complex cysts, and two of them were confirmed to be ovarian necrosis. In four patients who underwent IUA, the ovaries could be preserved. Conclusions: IUA is a promising therapy for preserving ovaries with simple cysts ≥40 mm in diameter; however, the indications for fetal surgery and the appropriate timing of intervention require further study.

## 1. Introduction

Fetal ovarian cysts are the most frequently diagnosed fetal intra-abdominal cysts. The incidence is approximately 1 in 2600 pregnancies [1]. The cause is not determined; however, a widely accepted mechanism is follicle production and maturation from fetal ovarian stimulation by fetal gonadotropins, maternal estrogens, and placental human chorionic gonadotropins [2]. Because fetal hypothalamus-pituitary-ovary maturation occurs after 29 weeks, most fetal ovarian cysts are diagnosed in the third trimester [3,4]. Maternal risk factors include diabetes mellitus, Rh isoimmunization, and preeclampsia, which are thought to be associated with excess fetal gonadotropins [5,6]. According to Nussbaum et al., fetal ovarian cysts have a diameter of ≥20 mm and are classified by ultrasonography as simple and complex cysts. Simple cysts are round, anechoic, unilocular, and thin-walled, while complex cysts are thick-walled with heterogeneous echogenicity, fluid-debris level, and intracystic septations (Figure 1) [7]. The presence of complex cysts strongly suggests ovarian torsion [8,9]. On ultrasonographical images, ovarian autoamputation, which may occur in the cases of ovarian ischemia following ovarian torsion, manifests as a freely mobile abdominal mass, the so-called wondering cyst, or wondering tumor [9,10]. Specifically, color Doppler imaging is employed to identify ovarian cysts, which are observed as masses without internal blood flow in the case of both simple and complex cysts. However, it is usually difficult to identify the ovarian artery.

The most serious complication of fetal ovarian cysts is ovarian torsion, followed by ovarian necrosis and adhesion formation. Polyhydramnios and ascites due to compression of the cyst against the intestinal tract have also been reported [11,12]. There has also been a report of a case of lung hypoplasia caused by a large fetal ovarian cyst [13]. The highest priority in perinatal management is careful follow-up to avoid ovarian torsion. In other words, monitoring the progression from simple to complex cysts is of utmost importance. However, the proper management of fetal ovarian cysts remains controversial. Although fetal ovarian cysts have been managed almost exclusively with expectant management, ovarian functional preservation should also be considered.

Prognostic factors for fetal ovarian cysts include cyst size and appearance [14]. In utero aspiration (IUA) can avoid fetal ovarian necrosis in simple cysts larger than 30–50 mm [11,15,16]. However, this is not generally indicated for complex cysts. However, there is little evidence of an established cutoff size and adequate gestational age for IUA and little discussion about the necessity and timing of intervention in complex fetal ovarian cysts. In this study, we retrospectively reviewed cases of prenatally diagnosed fetal ovarian cysts in our hospital, focusing on the relationship between cyst size and appearance, the timing of delivery, and prognosis. We also conducted a literature review on the diagnosis, management, fetal therapy, and delivery of cases of fetal ovarian cysts.

## 2. Materials and Methods

We retrospectively reviewed cases of fetal ovarian cysts diagnosed by prenatal ultrasonography at our institution between January 2010 and January 2020. The following were investigated: gestational age, cyst size, location, gestational age at maximum cyst size, maximum cyst diameter during pregnancy, last prenatal ultrasound findings, whether IUA was performed, progression of cyst size during pregnancy, and postpartum course. We classified fetal ovarian cysts as simple or complex according to the classification of Nussbaum et al. [7]. When fetal ovarian cysts were diagnosed, ovarian size and appearance were examined every 2 weeks. Before 2018, the perinatal management for all fetal ovarian cysts was expectant management, but after 2018, IUA was performed in cases of simple cysts >40 mm in diameter and <37 weeks of gestation (Figure 2).

The mode of delivery was generally vaginal delivery after the onset of labor except for cesarean section, which was an obstetrical consideration. In cases of fetal ovarian cysts >40 mm in diameter after 37 weeks of gestation, we induced labor to allow early surgical treatment to preserve ovarian function after 2018. After birth, the newborns underwent transabdominal ultrasonography by a pediatric surgeon, and those with simple cysts >40 mm in maximum diameter were treated surgically after informed consent was obtained from the parents. Complex cysts were managed conservatively without prenatal intervention. After birth, surgery was performed if the cyst was >40 mm in diameter or according to the parents’ wishes in the case of complex cysts.

### In Utero Aspiration

Patients were administered with prophylactic antibiotics and tocolysis with ritodrine preoperatively. Before the puncture, the operator confirmed the location of the placenta and cyst and selected a route that allowed access to the cyst and avoided placental puncture whenever possible. Perioperatively, IUA was performed under local anesthesia with xylocaine, and no fetal anesthesia was administered. A 21G percutaneous transhepatic cholangiography (PTC) needle with a length of 150 mm (Hakko, Japan) was used. The puncture was performed by two obstetricians with experience in fetal therapy. Various hormone levels (progesterone, estradiol, follicle-stimulating hormone, luteinizing hormone, and testosterone) of the aspired liquid were measured to confirm that the fluid was from an ovarian cyst. Postoperatively, we performed cardiotocography to confirm the fetus’s well-being, and follow-ups were scheduled every 2 weeks to monitor the cyst diameter and appearance.

## 3. Results

### 3.1. Outcome of Total Fetal Ovarian Cysts

In the present study, 36 cases of fetal ovarian cysts were diagnosed and managed using ultrasonography. The details of all cases are summarized in Table 1. Only Case 12 showed Rh isoimmunization. Of the 36 fetal ovarian cysts, 29 and 7 were simple and complex at diagnosis, respectively. Maternal and neonatal baseline characteristics, including maternal age, gestational age at diagnosis and delivery, maximum diameter, birth weight, simple cysts, and complex cysts, are summarized in Table 2. The outcomes of fetal ovarian cysts managed at our institution are shown in Figure 3.

### 3.2. Outcome of Simple Ovarian Cysts <40 mm in Diameter

Of the 29 patients with simple cysts, 14 had cysts <40 mm in diameter and two (Cases 12 and 14) progressed to complex cysts during prenatal follow-up and resolved spontaneously after birth. Of the 12 cysts that remained simple, one (Case 13) was surgically treated based on parents’ wishes, but the operated ovary was preserved. Of the remaining 11 cases, one (Case 1) was lost to follow-up, cysts in four cases (Cases 2, 4, 5, and 7) resolved spontaneously during pregnancy, and cysts in six cases (Cases 3, 6, and 8–11) resolved spontaneously after birth. None of the eight cases (Cases 2–9) of simple cysts <35 mm in diameter progressed to complex cysts requiring surgery after birth.

### 3.3. Outcome of Simple Ovarian Cysts ≥40 mm in Diameter

Fifteen of the simple cysts were >40 mm in diameter; of them, 11 were treated expectantly before 2018, while IUA was performed in four (Cases 15, 19, 26, and 29) after 2018. Cases 15 and 29 were treated with postnatal surgical treatment because the cyst diameter after birth was >40 mm. The median gestational age required for the IUA procedure was 34 weeks (range, 31–36 weeks). The hormone levels in cyst contents are summarized in Table 3.

Of the 11 cases of expectant management, cysts in three cases (Cases 18, 23, and 25) progressed to the complex cysts during prenatal follow-up. Of them, two (Cases 18 and 25) required postnatal surgical treatment, because the cyst diameter was >40 mm after birth, and both were confirmed to be ovarian necrosis. The cyst in Case 23 resolved spontaneously after birth. Five of the eight cases (Cases 16, 17, 22, 24, and 28) that remained simple cysts required postnatal surgical treatment, because the cyst diameter was >40 mm after birth; however, none were identified as ovarian necrosis or torsion. The remaining three cases (Cases 20, 21, and 27) did not require surgery, because the cyst diameter was <40 mm.

### 3.4. Outcome of Complex Ovarian Cysts

As for the seven cases of complex cysts, four (Cases 30–33) were <40 mm in diameter. Of these, the patient labelled as Case 31 was operated on after birth at the patient’s request. Cysts in three cases (Cases 34–36) were >40 mm in diameter, two of which (Cases 35 and 36) underwent postnatal surgical treatment, because the diameter was >40 mm after birth, both of which showed ovarian torsion and one of which (Case 36) showed ovarian necrosis. The cyst in Case 34 resolved spontaneously after birth, and the newborn was diagnosed with syndactyly after birth. The course of all infants after birth was good, with no reports of anemia or intestinal adhesions.

## 4. Discussion

Of the 36 prenatally diagnosed fetal ovarian cysts at our institution, 29 were simple cysts. Eight simple cysts <35 mm in diameter, except for Case 1, did not progress to complex cysts, and none required surgery after birth. Ovarian necrosis was observed in two of 11 (18%) patients who underwent expectant management for simple cysts ≥40 mm in diameter. In contrast, none of the four patients who underwent IUA for simple cysts >40 mm had no cases of necrosis. All cases in which ovarian necrosis was confirmed had ovarian cyst diameters of ≥40 mm and were diagnosed as complex cysts. Although many fetal ovarian cysts are diagnosed prenatally due to the improved performance of ultrasound equipment and the increased frequency of routine ultrasonography, consensus on perinatal management is lacking. The following are important clinical questions:

### 4.1. Diagnosis of Fetal Ovarian Cysts

Fetal ovarian cysts are almost always diagnosed in the third trimester [3,4]. Although the earliest diagnosis was reported at 19 weeks of gestation [17], fetal intra-abdominal cysts found before the third trimester are generally considered to be something other than ovarian cysts. Other differential diseases of fetal intra-abdominal cysts include simple renal cysts, multicystic dysplastic kidney, hydronephrosis, ureterocele, urachal cyst, hydrocolpos, enteric duplication cyst, mesenteric cyst, meconium pseudocyst, choledochal cyst, lymphangioma, and fetus in fetu [4]. In order to diagnose fetal ovarian cysts, external genitalia have to first be confirmed as being female organs. Subsequently, a cyst on the dorsal side of the bladder should be identified and the abovementioned differential diagnoses should be ruled out. However, if the cyst is large, it is often located in the midline, which makes it difficult to distinguish between the left and right sides. A characteristic finding of fetal ovarian cysts on ultrasonography is a daughter cyst, which is a small, round, and anechoic structure within the cyst (Figure 4). Lee et al. reported that this finding was present in 82% of ovarian cysts in neonates and infants, with a sensitivity, specificity, and positive predictive value of 82%, 100%, and 100%, respectively. This finding was not seen in other diseases that show intra-abdominal cysts (lymphangioma, enteric duplication cyst, enteric cyst, meconium pseudocyst, hydrometrocolpos, and urachal cyst) [18]. Similar findings have been reported in the fetus; however, the accuracy of these findings is unclear. Urachal, mesenteric, and enteric duplication cysts are quite indistinguishable, so the recognition of a daughter cyst seems to be exclusive to ovarian cysts. Additionally, there have been very few reports of malformations associated with fetal ovarian cysts, and the risk of chromosomal and non-chromosomal syndromes is very low [19]. However, Gaspari et al. reported that McCune-Albright syndrome (MAS) may be associated with fetal ovarian cysts [20]. Magnetic resonance imaging (MRI) is potentially useful in cases where ultrasonography is non-diagnostic, such as maternal obesity, fetal malposition, and oligohydramnios [4]. Furthermore, Nemec et al. reported that 2 of 16 cases that were diagnosed as simple cysts by ultrasonography were diagnosed as complex cysts after MRI. MRI provides more contrast than ultrasonography, allowing a more detailed characterization of the intracyst [21]. Although we did not make a definitive diagnosis of surgery in all of our cases after birth, the diagnostic accuracy was probably 100%, considering that the cysts that resolved spontaneously after birth were most likely ovarian cysts. There were no cases of postpartum malformations other than one case of syndactyly.

### 4.2. Perinatal Management of the Simple Cyst in Prenatal Diagnosis

Simple cysts can progress to complex cysts during pregnancy, which increases the risk of ovarian loss. Therefore, management of simple cysts is important. In their systematic review of the prognosis of 954 fetuses with fetal ovarian cysts, Bascietto et al. reported that diameter and appearance can help determine the prognosis. They also showed that fetal ovarian cysts >40 mm in diameter had a distinctly higher risk of ovarian torsion than cysts <40 mm (odds ratio, 30.8; 95% confidence interval, 8.6–110.0) [14]. In a systematic review of the prognosis of 365 fetuses with fetal ovarian cysts, Tyraskis et al. reported that fetal ovarian cysts measuring 30–59 mm in diameter had the highest risk (15–34%) of ovarian torsion [10]. All our cases that showed ovarian torsion had a maximum ovarian diameter of >50 mm, and all cases that showed ovarian necrosis had a maximum ovarian diameter of >40 mm.

A controversial issue in prenatal management of simple cysts is whether expectant management or IUA is more appropriate. Nakamura et al. reported that ovarian preservation was possible in 28 of 33 patients (85%) who underwent expectant management. They stated that the maximum ovarian diameter of all patients with ovarian torsion was >40 mm; therefore, expectant management may be appropriate otherwise [22]. Heling et al. reported that ovarian preservation was possible in 52 of 64 patients (81%), and that IUA should be performed only when vaginal delivery is difficult because of the cyst [19]. In contrast, there have been some reports that IUA has also been used to avoid ovarian torsion [9,15,16]. Many reports describe IUA performed at 40–50 mm or more, and it is effective for ovarian preservation. In a systematic review of 56 cases of IUA by Tyraskis et al., six (11%) cases showed ovarian torsion [10]. Only Diguisto et al. reported on the efficacy of IUA in a randomized controlled trial, with a significantly lower rate of oophorectomy (3%) after IUA, compared to 22% after expectant management of simple cysts of 30 mm or more [23]. Another advantage of IUA is that it can be used to diagnose fetal ovarian cysts by confirming the composition of the cyst contents. Lecarpentier et al. reported that the median estradiol level in the content fluid of fetal ovarian cysts was 12,500 ng/L (range, 1081–128,407 ng/L). They also reported that estradiol levels of 1000 ng/L or higher are diagnostic with 100% sensitivity and 100% specificity when compared to other diseases that cause intra-abdominal cysts (cloacal mass, urodigestive fistula mass, and urogenital sinus mass) [24]. Diguisto et al. also reported that the median estradiol level in the content fluid of fetal ovarian cysts was 12,694 ng/L (range, 2765–49,469 ng/L) [23]. The estradiol levels in our cases were above 1000 ng/L (range, 25,500–376,731 ng/L). The disadvantages of IUA are that reaccumulation occurs in 37.9% of cases, there is a possibility of amniotic fluid infection and preterm delivery, and nearly 10% of the procedures could not be performed, either because of fetal position or dry aspiration [10,14,23]. Various cutoff values have been proposed for the size of the ovary for IUA, including 30, 35, 40, and 50 mm [15,23,25,26]. We encountered no cases of ovarian torsion or ovarian necrosis in patients <35 mm; however, further studies are needed. Because fetal ovarian cysts are generally not fatal, it remains controversial whether fetal therapy should be indicated. Although the preservation of ovarian function is an important issue, the reproductive rate after unilateral ovarian surgery in young women is reported to be 60–80%, even after chemotherapy for early-stage malignancy; thus, some fertility remains [27]. Further discussion is needed on whether IUA is appropriate, considering the balance between the risk of preterm birth and fetal death (even if the frequency is quite low) and the benefits of preserving ovarian function. However, there is still little debate regarding when IUA should be performed.

### 4.3. Perinatal Management of the Complex Cyst in Prenatal Diagnosis

Complex cysts are strongly suggestive of ovarian torsion, and Bascietto et al. reported in a systematic review that 44.9% of prenatally diagnosed complex cysts had ovarian torsion [14]. Patients with prenatally diagnosed complex cysts or cysts progressing to complex require careful follow-up from the prenatal to postnatal period. This is because there have been reports of cases of fetal anemia due to intracystic hemorrhage and cases of intestinal obstruction after birth [25,28,29]. Additionally, the possibility of a tumor must be considered, because there have been cases of complex cysts from the time of first diagnosis that were diagnosed as cystadenoma or granulosa cell tumor after birth [30,31]. However, there was only one case of malignancy reported in a newborn; thus, the possibility of malignancy was considered to be extremely low [32]. The problem with managing complex cysts is that, despite their classification, they do not always lead to ovarian necrosis. In our case, there were three cases of necrosis in five cases with complex cysts operated after birth. Interestingly, all cases of necrosis had fluid-debris level. None of the cases with echogenicity or separation showed necrosis. In fact, Ozcan et al. reported that a complex cyst with a fluid-debris level is particularly suggestive of ovarian torsion [9]. However, the limitation of this study is that we did not confirm intraperitoneal findings in all cases. Currently, IUA is not indicated for the treatment of complex cysts because of the high likelihood of ovarian torsion. Rather, this intervention should mostly be limited to simple cysts. Even in case IUA becomes a recommended treatment for complex cysts in the future, it will likely remain discouraged for complex cysts with fluid-debris level. Current evidence suggests that when these are diagnosed prenatally, it is appropriate to perform ultrasonography longitudinally to follow up on ovary size and middle cerebral artery-peak systolic velocity to investigate fetal anemia.

### 4.4. Timing and Mode of Delivery for Fetal Ovarian Cysts

Although proper perinatal management is controversial, many authors recommend spontaneous vaginal delivery after the onset of labor at full term with careful ultrasound rather than iatrogenic preterm birth or cesarean section. This is because fetal ovarian cysts are not fatal [4,14]. The biggest question is whether termination should be performed to preserve ovarian function. Early termination of pregnancy can lead to early postnatal treatment. In contrast, because fetal ovarian cysts sometimes shrink spontaneously during pregnancy, early termination of pregnancy can increase the number of unnecessary surgeries. In our institution, there were cases of termination of pregnancy for fetal ovarian cysts >40 mm in diameter after 37 weeks of gestation; however, it was unclear whether this may have contributed to preservation of ovarian function, and few studies have discussed this point. In our study, Cases 17, 24, and 28, the maximum diameter of the ovarian cyst was >40 mm in diameter, and labor was induced at 37–39 weeks to avoid ovarian torsion; however, ovarian torsion was observed in Case 24. In Cases 16, 20, 21, 22, and 27, we did not induce labor. With the exception of Case 16, the remaining cases were delivered after 39 weeks, but no ovarian torsion was observed. We did not find any association between early pregnancy termination and preservation of ovarian function in our cases. Preterm delivery may be considered in cases of severe fetal anemia due to intracystic hemorrhage or bilateral ovarian cysts >40 mm in diameter. Elective cesarean section should also be considered in cases in which vaginal delivery is expected to be difficult due to massive fetal ovarian cysts. Because fetal ovarian cysts may remain after delivery and surgery may be necessary, delivery should be performed at a perinatal center. We propose diagnostic guidelines for the identification of fetal ovarian cysts, as shown in Figure 5.

## 5. Conclusions

We encountered 36 cases of fetal ovarian cysts at our institution. In the case of a simple cyst with a cyst diameter of 35 mm, there was no ovarian torsion or ovarian necrosis. There were no cases of ovarian torsion or necrosis in patients who underwent IUA. However, the indications and timing of IUA need to be further investigated. Of the 36 cases, 3 had ovarian necrosis, all of which were complex cysts with fluid-debris level. In addition, the prenatal management of complex cysts needs to be further investigated because of a lack of evidence.

## Figures and Tables

**Figure 1 diagnostics-11-02224-f001:**
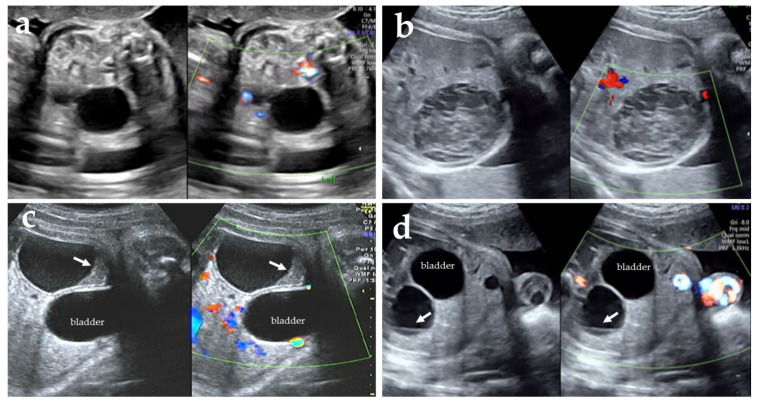
Typical ultrasonographic images of fetal ovarian cysts with grayscale and color Doppler mode. (**a**) A simple completely anechoic cyst, (**b**) A complex cyst with heterogenous echogenicity, (**c**) A complex cyst with fluid-debris level (arrows), (**d**) A complex cyst with a single septation (arrows).

**Figure 2 diagnostics-11-02224-f002:**
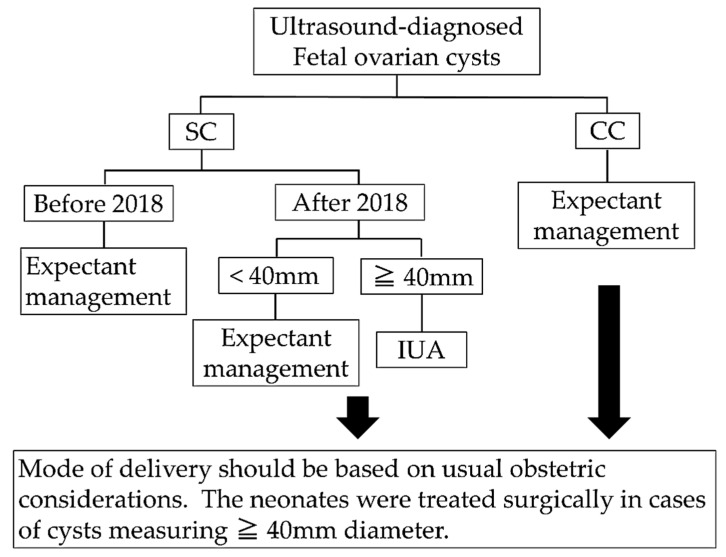
Prenatal management policy of fetal ovarian cysts in our institution. CC, complex cyst; IUA, in utero aspiration; SC, simple cyst.

**Figure 3 diagnostics-11-02224-f003:**
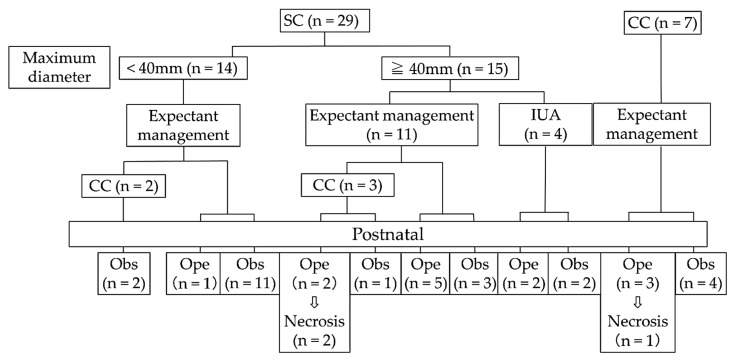
Outcomes of fetal ovarian cysts managed in our institution. CC, complex cyst; IUA, in utero aspiration; Ope, operation; Obs, observation; SC, simple cyst.

**Figure 4 diagnostics-11-02224-f004:**
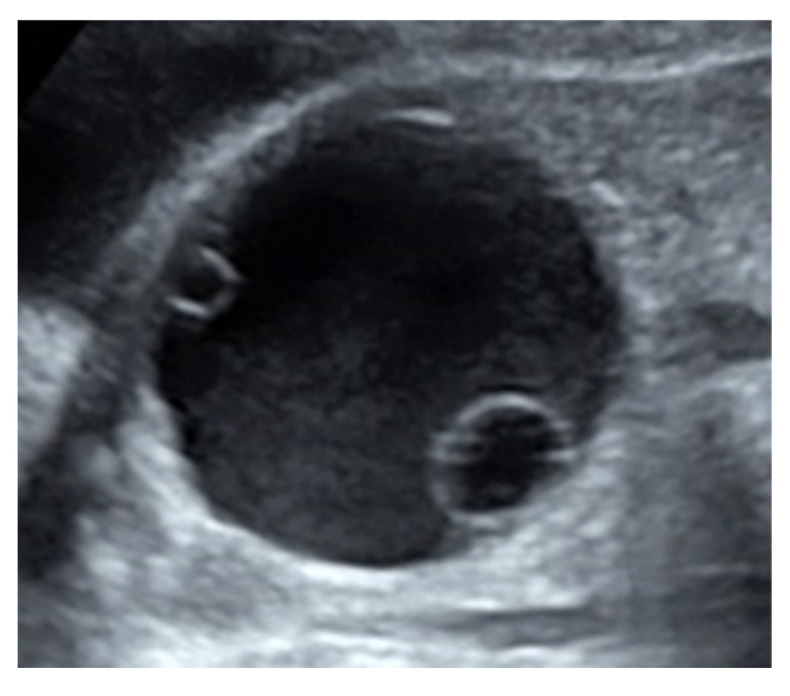
Daughter cyst. The specific imaging features are exclusively observed in fetal ovarian cysts, but not in other intra-abdominal cyst diseases.

**Figure 5 diagnostics-11-02224-f005:**
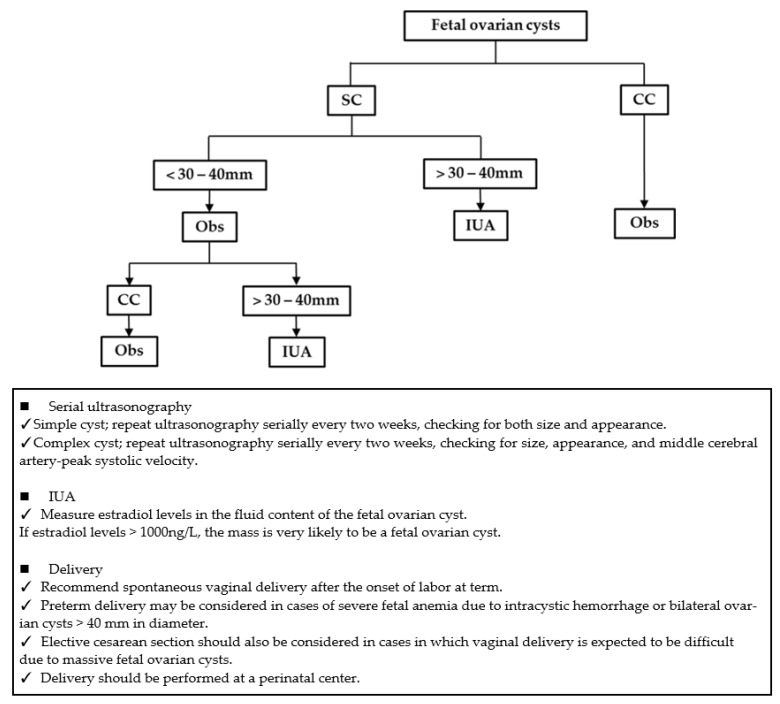
Diagnostic guidelines for the identification of fetal ovarian cyst. CC, complex cyst; IUA, in utero aspiration; Obs, Observation; SC, simple cyst.

**Table 1 diagnostics-11-02224-t001:** Prenatal and postnatal course of 36 cases prenatally diagnosed fetal ovarian cyst.

Prenatal												Postnatal					
Case No.	GA at Diagnosis (Weeks)	Ultrasonographic Features at Diagnosis	Side	Size at Diagnosis(mm)	GA at Maximum Diameter (Weeks)	Maximum Diameter (mm)	Ultrasonographic Features at Last Prenatal Scan	GA Changed from SC to CC	IUA (weeks)	Size Progression in Pregnancy	GA at Delivery	Ultrasonographic Features at Postnatal Scan	Treatment	Age at Surgery	Torsion	Necrosis	Pathology
1	29	Simple	R	13	30	14	Simple			Reduced			Lost follow				
2	35	Simple	R	18	35	18	Simple			Reduced	38	not visible	S.R				
3	31	Simple	L	26	31	26	Simple			Reduced	38	Simple	S.R				
4	32	Simple	R	28	32	28	Simple			Reduced	40	not visible	S.R				
5	32	Simple	R	28	32	28	Simple			Reduced	40	not visible	S.R				
6	31	Simple	R	29	31	29	Simple			Reduced	40	Simple	S.R				
7	27	Simple	R	27	29	29	Simple			Reduced	41	not visible	S.R				
8	30	Simple	L	26	33	29	Simple			Reduced	39	Simple	S.R				
9	32	Simple	R	31	32	31	Simple			Reduced	40	Simple	S.R				
10	33	Simple	L	35	33	35	Simple			Reduced	38	Simple	S.R				
11	32	Simple	L	22	34	35	Simple			Increased	37	Simple	S.R				
12	34	Simple	L	36	34	36	Complex (heterogeneous echogenicity)	37		Reduced	40	Complex (heterogenous echogenicity)	S.R				
13	33	Simple	L	37	33	37	Simple			Reduced	39	Simple	LF	0 day	No	No	
14	32	Simple	L	39	32	39	Complex (septation)	36		Reduced	40	Complex (septation)	S.R				
15	33	Simple	R	42	33	42	Simple		33	Increased	37	Simple	LAAC	0 day	No	No	Simple cyst
16	32	Simple	R	30	36	42	Simple			Increased	37	Simple	LF	1 M	No	No	
17	35	Simple	L	44	35	44	Simple			Reduced	37	Simple	LF	0 day	No	No	
18	30	Simple	R	30	36	46	Complex (fluid-debris level)	36		Increased	40	Complex (fluid-debris level)	LAAC	0 day	No	Yes	Simple cyst
19	33	Simple	L	48	33	48	Complex (heterogeneous echogenicity)	34	33	Reduced	39	not visible	S.R				
20	29	Simple	L	20	35	49	Simple			Increased	41	Simple	S.R				
21	34	Simple	R	23	36	49	Simple			Increased	39	Simple	S.R				
22	32	Simple	L	46	36	52	Simple			Increased	39	Simple	LF	0 day	No	No	
23	31	Simple	R	53	31	53	Complex (septation)	33		Reduced	39	Complex (septation)	S.R				
24	34	Simple	L	50	37	53	Simple			No change	38	Simple	LF	1 day	Yes	No	
25	34	Simple	R	40	37	60	Complex (fluid-debris level)	37		No change	38	Complex (fluid-debris levels)	LF	1 day	Yes	Yes	
26	30	Simple	L	50	31	61	Complex (heterogeneous echogenicity)	32	31	Reduced	39	not visible	S.R				
27	33	Simple	L	40	37	65	Simple			Reduced	40	Simple	S.R				
28	31	Simple	L	41	37	69	Simple			Reduced	38	Simple	OF	0 day	No	No	
29	35	Simple	L	81	35	81	Simple		36	Reduced	37	Simple	LF	2 day	No	No	
30	35	Complex (septation)	L	20	35	20	Complex (septation)			Reduced	38	not visible	S.R				
31	37	Complex (fluid-debris level)	R	26	38	30	Complex (fluid-debris level)			Increased	39	Complex (fluid-debris level)	LAAC	0 day	No	No	simple cyst
32	30	Complex (septation)	R	27	33	32	Complex (septation)			No change	39	Complex (septation)	S.R				
33	34	Complex (septation)	R	32	37	35	Complex (heterogeneous echogenicity)			Reduced	39	not visible	S.R				
34	31	Complex (septation)	L	54	31	54	Complex (fluid-debris level)			Reduced	37	Complex	S.R				
35	35	Complex (septation)	R	35	36	55	Complex (septation)			Increased	37	Complex	LF	0 day	No	No	
36	30	Complex (fluid-debris level)	L	55	35	60	Complex (fluid-debris level)			No change	39	Complex (fluid-debris levels)	LAAC	4 day	Yes	Yes	Simple cyst or Serous cystadenoma

CC, complex cyst; GA, gestational age; IUA, in utero aspiration; L, left side; LAAC, laparoscopic assisted abdominal cystectomy; LF, laparoscopic fenestration; OF, open fenestration; R, right side; SC, simple cyst; SR, spontaneous regression.

**Table 2 diagnostics-11-02224-t002:** Maternal and neonatal baseline characteristics.

	Total (*n* = 36)	SC (*n* = 29)	CC (*n* = 7)
Maternal age (years)	30 (23–42)	30 (23–42)	29 (24–37)
Gestational age at diagnosis (weeks)	32 (27–37)	32 (27–37)	34 (30–37)
Maximum diameter (mm)	39 (22–79)	39 (22–79)	32 (22–58)
Gestational age at delivery (weeks)	39 (37–41)	39 (37–41)	39 (37–39)
Birth weight (g)	2914 (2154–3758)	2917 (2154–3758)	2914 (2546–3630)

Data are presented as median (range). CC, complex cyst; SC, simple cyst.

**Table 3 diagnostics-11-02224-t003:** Hormone levels of aspiration liquid.

Hormone Levels of Aspiration Liquid	Case 15	Case 19	Case 26	Case 29
Progesterone (ng/mL)	3390	2940	1330	1250
Estradiol (pg/mL)	378,205	376,731	737,443	25,500
FSH (mIU/mL)	0.6	0.5	0.2	
LH (mIU/mL)	0.1	0.1	<0.1	
Testosterone (ng/mL)	5.48	2.79	3.99	

FSH, follicle stimulating hormone; LH, luteinizing hormone.

## Data Availability

The data presented in this study are available upon request from the corresponding author.

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
