# Peer review of "Diagnosis, Management, and Therapy of Fetal Ovarian Cysts Detected by Prenatal Ultrasonography: A Report of 36 Cases and Literature Review"

_diagnostics, 2021, doi:10.3390/diagnostics11122224_

Round 1

Reviewer 1 Report

This is a good cohort and literature review but the authors need to 'take a stand' and provide an algorithm for best practice based on their findings (US and hormone levels) and review.

Another questions is the US procedure used trans-abdominal vs vaginal in an attempt to get the best view of the cyst characteristics.

Need to indicate if the fluid-debris levels are present that necrosis has occurred and prenatal procedures will have no value as the IUA is to prevent this important female outcome.

There was no comment that fetal genitalia imaging was used in the process to validate ovarian pathology.

Author Response

First, we would like to thank the editors and reviewers for handling our manuscript. We appreciate for raising important issues on our manuscript and believe our revised manuscript is now much improved by addressing those points.

Reviewer 1

Comments and Suggestions for Authors

This is a good cohort and literature review but the authors need to 'take a stand' and provide an algorithm for best practice based on their findings (US and hormone levels) and review.

Thank you for your suggestions. We have created a new Figure 5 in the Discussion section, an algorithm for the best management of fetal ovarian cysts.

Another questions is the US procedure used trans-abdominal vs vaginal in an attempt to get the best view of the cyst characteristics.

We thank the reviewer to point this out. The diagnosis of fetal ovarian cysts is usually made by transabdominal ultrasound and not by transvaginal ultrasound. This is because transvaginal ultrasonography cannot observe the abdominal organs of the fetus, especially when evaluating a fetus after the second trimester. Therefore, only cases of fetal ovarian cysts diagnosed by transabdominal ultrasonography are included in this study.

Need to indicate if the fluid-debris levels are present that necrosis has occurred and prenatal procedures will have no value as the IUA is to prevent this important female outcome.

We thank the reviewer to point this out. As the reviewer pointed out, a complex cyst is not an indication for IUA because ovarian torsion (and necrosis) is likely to have occurred. Additionally, a complex cyst with fluid-debris levels is included in the complex cyst. This is described in Lines 44-45. It has been reported that complex cysts, especially those with fluid-debris levels, are strongly associated with ovarian necrosis. This is discussed in Lines 344-346. Therefore, IUA is not indicated for cases of complex cysts, and IUA is performed only for cases of simple cysts. This content has been added to Line 70. To avoid confusing to readers, we have changed the content to the following (Line348-351).

“Currently, IUA is not indicated for the treatment of complex cysts because of the high likelihood of ovarian torsion. Rather, this intervention should mostly be limited to simple cysts. Even in case IUA becomes a recommended treatment for complex cysts in the future, it will likely remain discouraged for complex cysts with fluid-debris levels.”

There was no comment that fetal genitalia imaging was used in the process to validate ovarian pathology.

We thank the reviewer to point this out. We have added the following to the Discussion section.

“In order to diagnose fetal ovarian cysts, external genitalia have to be first confirmed as being female organs.”

Thank you again for your thoughtful comments.

Reviewer 2 Report

In this study authors retrospectively reviewed and analyzed the clinical significance of their cases of fetal ovarian cysts diagnosed by prenatal ultrasonography.The subject is of interest and the analysis well performed so I would like to congratulate with authors for their effort.

My suggestions are:

1)I will add a ultra sonographic image showing the presence or absence of Color Doppler

2) I will add the ultra sonographic description of signs of ovarian torsion.

3) Line 96 please describe PTC

4) Describe the risk of chromosomal anomalies and nonchromosomal syndromes (very low) when an ovarian cyst is present

5) Better describe differential diagnosis (urachal,mesenteric and enteric duplication cysts are quite indistinguishable, so the recognition of a daughter cyst seems to be exclusive to ovarian cysts).

Author Response

First, we would like to thank the editors and reviewers for handling our manuscript. We appreciate for raising important issues on our manuscript and believe our revised manuscript is now much improved by addressing those points.

Reviewer 2

Comments and Suggestions for Authors

In this study authors retrospectively reviewed and analyzed the clinical significance of their cases of fetal ovarian cysts diagnosed by prenatal ultrasonography. The subject is of interest and the analysis well performed so I would like to congratulate with authors for their effort.

My suggestions are:

1)I will add a ultrasonographic image showing the presence or absence of Color Doppler

We thank the reviewer to point this out. We have included a color Doppler imaging in Figure 1. We also added the following explanation.

“Specifically, color Doppler imaging is employed to identify ovarian cysts, which are observed as masses without internal blood flow in the case of both simple and complex cysts. However, it is usually difficult to identify the ovarian artery.”

2) I will add the ultrasonographic description of signs of ovarian torsion.

We thank the reviewer to point this out. As we described in Line43-45, complex cysts are findings that strongly suggest ovarian torsion. It is difficult to determine the presence or absence of blood flow to the ovaries or the findings that directly indicate the ovarian torsion as in adults.

We also added the following explanation in Line47-52.

“On ultrasonographical images, ovarian autoamputation, which may occur in the cases of ovarian ischemia following ovarian torsion, manifests as a freely mobile abdominal mass, the so-called wondering cyst, or wondering tumor [9,10]. Specifically, color Dop-pler imaging is employed to identify ovarian cysts, which are observed as masses without internal blood flow in the case of both simple and complex cysts. However, it is usually difficult to identify the ovarian artery.”

3) Line 96 please describe PTC

We apologize for the inadequate explanation. PTC needle means “percutaneous transhepatic cholangiography” needle. The full spelling has been added to the revised manuscript.

4) Describe the risk of chromosomal anomalies and nonchromosomal syndromes (very low) when an ovarian cyst is present

We thank the reviewer to point this out. The following was added to the Discussion section.

“the risk of chromosomal and non-chromosomal syndromes is very low [19]. However, Gaspari et al. reported that McCune-Albright syndrome (MAS) may be associated with fetal ovarian cysts [20].”

5) Better describe differential diagnosis (urachal, mesenteric and enteric duplication cysts are quite indistinguishable, so the recognition of a daughter cyst seems to be exclusive to ovarian cysts).

We thank the reviewer to point this out. We have added the above sentence and following to the Discussion section.

“In order to diagnose fetal ovarian cysts, external genitalia have to be first confirmed as being female organs. Subsequently, a cyst on the dorsal side of the bladder should be identified and the abovementioned differential diagnoses should be ruled out. How-ever, if the cyst is large, it is often located in the midline, which makes it difficult to distinguish between the left and right sides.”

We have also added an image of the daughter cyst as Figure 4.

Thank you again for your thoughtful comments.

Round 2

Reviewer 1 Report

Thank you for this revision and it appears that all the concerns were answered.